# Nexus between constructs of social cognitive theory model and diabetes self-management among Ghanaian diabetic patients: A mediation modelling approach

Yaa Obirikorang[1], Emmanuel Acheampong[2,3,4]*, Enoch Odame Anto[3,4,5], Ebenezer Afrifa-Yamoah[6], Eric Adua[4], John Taylor[3,4], Linda Ahenkorah Fondjo[2], Sylvester Yao Lokpo[7], Evans Asamoah Adu[2], Bernard Adutwum[1], Enoch Ofori Antwi[1], Emmanuella Nsenbah Acheampong[2], Michael Adu Gyamfi[7], Freeman Aidoo[2], Eddie-Williams Owiredu[2], Christian Obirikorang[2]

1 Department of Nursing, Faculty of Health Sciences, Garden City University College (GCUC), Kenyasi, Kumasi, Ghana, 2 Department of Molecular Medicine, School of Medical Science, Kwame Nkrumah University of Science and Technology (KNUST), Kumasi, Ghana, 3 Centre for Precision Health, ECU Strategic Research Centre, Edith Cowan University, Perth, Australia, 4 School of Medical and Health Sciences, Edith Cowan University, Joondalup, Australia, 5 Department of Medical Diagnostics, College of Health Sciences, Kwame Nkrumah University of Science and Technology, Kumasi, Ghana, 6 School of Science, Edith Cowan University, Joondalup, Australia, 7 Department of Medical Laboratory Sciences, School of Allied Health Sciences, University of Health and Allied Sciences, Ho, Ghana

* emmanuelachea1990@yahoo.com

## Abstract

The promotion of Diabetes Self-Management (DSM) practices, education, and support is vital to improving the care and wellbeing of diabetic patients. Identifying factors that affect DSM behaviours may be useful to promote healthy living among these patients. The study assessed the determinants of DSM practices among Type 2 diabetes mellitus (T2DM) patients using a model-based social cognitive theory (SCT). This cross-sectional study comprised 420 (T2DM) patients who visited the Diabetic Clinic of the Komfo Anokye Teaching Hospital (KATH), Kumasi-Ghana. Data was collected using self-structured questionnaires to obtain socio-demographic characteristics, T2DM-related knowledge, DSM practices, SCT constructs; beliefs in treatment effectiveness, level of self-efficacy, perceived family support, and healthcare provider-patient communication. Path analysis was used to determine direct and indirect effects of T2DM-related knowledge, perceived family support, and healthcare provider service on DSM practices with level of self-efficacy mediating the relationships, and beliefs in treatment effectiveness as moderators. The mean age of the participants was 53.1(SD = 11.4) years and the average disease duration of T2DM was 10 years. Most of the participants (65.5%) had high (>6.1mmol/L) fasting blood glucose (FBG) with an average of 6.93(SD = 2.41). The path analysis model revealed that age ($p = 0.176$), gender ($p = 0.901$), and duration of T2DM ($p = 0.119$) did not confound the relationships between the SCT constructs and DSM specified in the model. A significant direct positive effect of family and friends' support (*Critical ratio (CR) = 5.279, p < 0.001*) on DSM was observed. Self-efficacy was a significant mediator in this relationship (*CR = 4.833, p < 0.001*). There

**Data Availability Statement:** SPSS file of the dataset, on which conclusions of this paper were made, is available as a Supporting Information file.

**Funding:** The authors received no specific funding for this work.

**Competing interests:** The authors have declared that no competing interests exist.

were significant conditional indirect effects (CIE) for knowledge of T2DM and family and friends' support at medium and high levels of belief in treatment effectiveness *(p < 0.05)* via level of self-efficacy on DSM practices. However, no evidence of moderated-mediation was observed for the exogenous variables on DSM. Diabetes-related knowledge of T2DM, family and friends' support, level of self-efficacy, and belief in treatment effectiveness are crucial in DSM practices among Ghanaian T2DM patients. It is incumbent to consider these factors when designing interventions to improve DSM adherence.

## Background

Diabetes mellitus (DM) is a chronic health condition with devastating consequences on patients and public health. It is characterized by high blood glucose levels that are caused by defects in insulin release or action [1]. Type II Diabetes mellitus (T2DM), constitute over half of all known diabetes cases, affecting an estimated 463 million adults globally [2, 3]. Meanwhile, the International Diabetes Federation (IDF) has extrapolated that 700 million people will have the disease by 2045 [2, 3]. The most disturbing aspect is that individuals in sub-Saharan Africa are the most affected, with a recorded prevalence ranging between 7–20% [4]. Currently, Ghana's diabetes prevalence is 6.4% [5] in adults and if measures to confront the disease are not revamped, the prevalence is likely to increase due to the country's already under-resourced and outstretched healthcare system and lifestyle behaviours.

Research has shown that T2DM progression is due to poor lifestyle choices and hence, promoting self-management practice through the modification of health behaviours, is the surest way to delaying diabetes-related mortality and morbidity [6, 7]. Basic diabetes self-management (DSM) practices include medication adherence, daily glucose monitoring, maintaining a healthy diet, regular exercising, and daily foot examinations [8–10]. Accumulating evidence indicates that compliance and implementation of DSM practices improve the health of T2DM patients, whereas T2DM patients with little self-management skills are prone to the negative consequences of the disease [11, 12].

To this end, researchers and health professionals have developed theories and cognitive models in pursuit of health promotion behaviours and interventions [13, 14]. However, most of the health promotion and theories only predict health behaviour and cannot explain the interactions between the constructs of the models [13, 15]. Social cognitive theory (SCT) has emerged as the most suitable model for examining health-related behaviours among individuals with chronic diseases [12, 16]. SCT proposes that cognitive processes could develop someone's behaviour. The theory explains health-associated behaviours based on a three-way mutual interaction between environmental factors, personal factors, and behaviours [12, 16, 17].

Several studies have reported on the SCT model in forecasting self-care among T2DM patients and demonstrated that the actions of patients with T2DM could be influenced by their self-care regime [14, 18]. SCT has been reported to have several constructs such as knowledge of diabetes, self-efficacy, and belief in treatment effectiveness family and friends' support, healthcare-patient communication [19–21]. These constructs interact with each other to allow patients to retain the influence of their disease [22]. Therefore, the SCT model provides opportunities to examine and explore the interaction between personal and environmental factors that influence DSM behaviours [12].

In Ghana, self-management practices of T2DM include healthy eating, being active and doing regular aerobic exercise, regular blood glucose monitoring, medical adherence, and

knowledge about general complications of uncontrolled diabetes [23, 24]. Therefore, studies conducted among T2DM patients in Ghana have focused on medical compliance [25], factors that affect patients' compliance to self-care activities [26], and a combination of both medication adherence and self-care behaviours [24]. While findings from studies have been impactful, they also revealed a lack of studies of the psychological aspect of self-management among Ghanaian T2DM patients. Moreover, these studies used correlation analyses for data collection that do not provide adequate information about the interactivity between independent and dependent variables.

SCT model is very effective in predicting and explaining DSM, however, most of the studies have been done in advanced countries Hence, the present study employed the SCT model to determine predictors for proper DSM practices among T2DM patients in a Ghanaian population.

## Methods

### Study design/area

The study was a hospital-based cross-sectional study design undertaken from November 2018 to April 2019 at the Diabetes Clinic of Komfo Anokye Teaching Hospital (KATH) in the Ashanti Region of Ghana, Kumasi. KATH is in Kumasi, the Regional Capital of the Ashanti Region with a total projected population of 4,780,380 (2010 Ghana Population Census). KATH is the second-largest hospital in Ghana and Diabetes Clinic is part of the medical directorate that provides services to about 250 patients from Monday to Friday per week with an average of 10,706 out-patient attendance per year. The geographical location of the thousand two hundred- (1200-) bed capacity, the road network of the country, and commercial nature of Kumasi makes the hospital accessible to all the areas that share the boundaries with Ashanti Region and others that are further away. KATH takes direct referrals from 12 out of the 16 administrative regions in Ghana. These are the Ashanti, Bono, Bono East, Ahafo, Western North, Savannah, Northern, Northeast, Upper East, Upper West, and some parts of the Central and Eastern regions of Ghana. It also receives patients from neighbouring countries such as Ivory Coast and Burkina Faso. The diabetic centre of the KATH is situated beneath the medicine block (D block) just between the chest clinic and diagnostic centre and behind the emergency unit of the hospital.

### Study population and subject sampling strategy

A consecutive sampling approach was used to recruit a total of 420 T2DM participants aged 30 years and above who visited the KATH Diabetic Clinic for routine check-ups and treatment. Participants were consecutively recruited until the calculated sample size was achieved. This study was conducted in consultation with clinicians and qualified health professionals. T2DM was diagnosed by clinicians at KATH, and it was established based on the international classification of disease (ICD-10-CM Diagnosis Code E11.9). Each patient was carefully examined, and their medical records were thoroughly reviewed. As a result, we excluded all those individuals who were suffering from cancer, arthritis, infectious diseases, cardiovascular disease, thyroid disorders, pituitary disorders, and adrenal disorders. The study did not include pregnant and lactating mothers. Since T2DM is largely a disease of ageing, the study recruited only individuals who were 30 years and above. Furthermore, to limit potential confounding and the likelihood of recruiting participants with type 1 diabetes, we excluded participants on insulin injections. Patients who were physically or mentally challenged and those who had less than 6 months duration of diabetes were excluded.

## Sample size justification

Using the standard normal variate for significance (Z) of 1.96, 5% margin of error (d), and an assumed adherence rate (p) of 39.2% to self-care behaviours among adult T2DM patients [27], the recommended minimum sample size (n) for the study was 366 using the formula $n = [p(1-p)] \times z^2/d^2$. However, to adjust for random error and strong statistical power, a maximum sample of 420 was used for the study.

## Ethical consideration

Ethical approval for the study was obtained from the Committee on Human Research, Publication, and Ethics of the School of Medicine and Dentistry, Kwame Nkrumah University of Science and Technology (KNUST), and KATH Ethical Committee (ref: CHRPE/AP/084/17). Participation was voluntary and written informed consent was obtained from each participant according to the Helsinki Declaration.

## Conceptual framework and hypotheses

The objective of self-management strategies is to assist patients with chronic conditions to perform therapeutic and preventive health-associated activities. Self-management behaviour is the mainstay of diabetes care. As a process-oriented approach, SCT epitomises direct and indirect analysis of the relationship between personal and environmental factors, and DSM. Personal factors involve diabetes-related knowledge, belief in treatment effectiveness, self-efficacy while environmental factors include family and friends support, and healthcare providers' support [18].

Knowledge of the basic physiology of diabetes, medication, diet, testing, and monitoring of blood sugar, general diabetes are important elements of diabetes management [28]. Knowledge of diabetes has been shown to be associated with DSM but exerted influence on DSM among T2DM individuals through belief in treatment effectiveness and self-efficacy [29]. It is evident in the literature that belief in treatment effectiveness is associated with positive DSM among T2DM patients [30–32]. Interestingly, belief in treatment effectiveness seems to moderate relationships between diabetic knowledge, provider-patient communication, family support, and DSM among T2DM patients.

Self-efficacy implies a person's confidence to carry out healthy behaviour. It is one of the principal concepts of SCT that is fundamental to behavioural accomplishment [33, 34]. Besides the exploration of the association between personal factors and DSM among T2DM individuals, studies have shown that family and friend support exerts its effect on DSM through belief in treatment effectiveness and self-efficacy [34–36]. These pieces of evidence show that family and friends' support could directly or indirectly impact DSM among T2DM. Communication between a healthcare provider and a patient is critical in the management of diabetes. Studies have reported that the absence of advice from healthcare providers negatively affects DSM among T2DM [11, 34, 37].

Therefore, we hypothesized that self-efficacy mediates the relationship between personal and environmental factors and DSM with belief in treatment effectiveness moderating this association [**Fig 1**].

## Measurement instrument

A self-reported questionnaire and validated questionnaires were used for the data collection on study variables. The first part of the questionnaire consisted of 28 questions divided into three sections including section A: demographic characteristics, section B: health profile, and

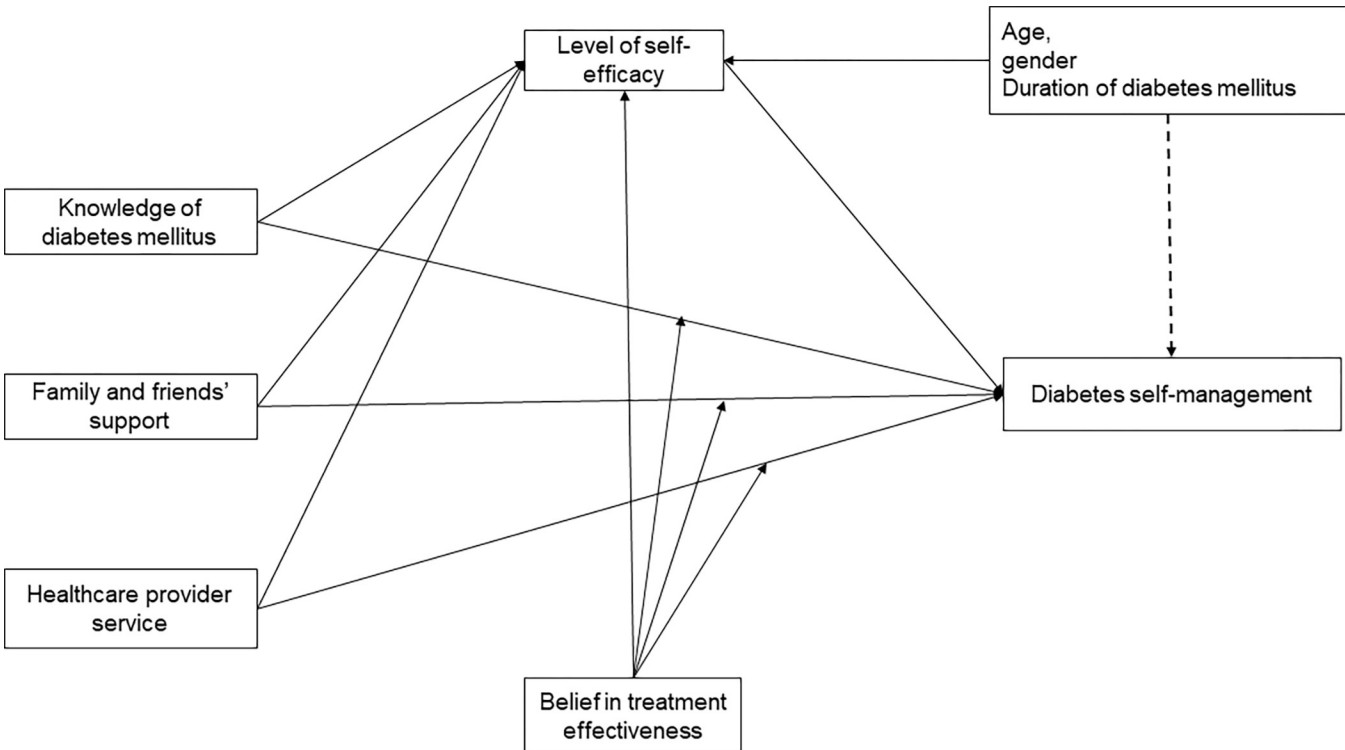

**Fig 1. Hypothesized model of determinants influencing DSM among T2DM in a Ghanaian population.** Fig 1 shows modified model functions of predictors of DSM. The level of self-efficacy mediates the relationship of personal and environmental factors with DSM. The dependent variables include diabetes-related knowledge, belief in treatment effectiveness, self-efficacy, family and friends' support, and healthcare provider service. The dependent variable is diabetes self-management.

section C: knowledge of T2DM. The second part consisted of 39 validated questions that measured DSM practices and SCT constructs. The DSM practices and SCT questionnaires have been validated [11, 12, 37, 38] respectively. In addition, we assessed the reliability of the questionnaires in our setting as described in the sections below. The questionnaire was completed by each patient via a face-to-face interview approach by ethically recognised researchers and in consultation with clinicians. All interviews and data were collected at the Diabetic Clinic of the KATH. Interviews were conducted during the morning hours for the regular diabetic Clinic at the KATH.

## Measurement of DSM

DSM practices were assessed using the summarised version of Diabetes Self-Care Activities (DSCA) [37, 38]. The study utilised a revised DSCA constituting 10 items that determine medical adherence (2 items), healthy eating habits (2 items), physical activities (2 items) foot care (2 items), and FBG testing (2 items) [38]. Each item had a scale response that ranged from 1 to 7 indicating diabetes self-care practices over one week. The total score on the scale ranged between 0 and 70. Internal consistency and reliability of the scale were assessed to be α = 0.744.

## Level of self-efficacy

Self-efficacy for diabetes scale was used to measure perceived self-efficacy [33, 34]. Seven items were used to measure the respondents' perceived self-efficacy. On a Likert scale of 0 (definitely,

yes) to 4 (definitely not) where higher scores indicate poorly perceived self-efficacy, each item was reversed coded to indicate higher scores for higher perceived self-efficacy. Internal consistency and reliability of the scale were assessed to be $\alpha = 0.618$.

### Belief in treatment effectiveness

To measure belief in treatment effectiveness, we adapted a questionnaire from Xu, [37]. The scale of the questionnaire was defined as the perceived relevance of self-care in managing diabetes. Eight items were used to assess the perceived benefits. Each item had a Likert scale that ranged from 0 (Not important) to 4 (extremely important) to measure the grades of perceived benefits. Internal consistency and reliability of the scale were assessed to be $\alpha = 0.746$.

### Family and friends' support

The scale used to measure family support was adapted from Xu et al [11]. The scale consisted of 7 items for measuring family supportive behaviours on a Likert scale of "never = 0" to "always = 4". The reliability and validity of the family support scale were determined to be $\alpha = 0.827$.

### Healthcare provider service

Seven questions were used to measure healthcare provider-patient communication. Item scale ranged from "never = 0" to "always = 4". Internal consistency and reliability of the scale were assessed to be $\alpha = 0.496$. The scale used to measure healthcare provider-patient communication was adapted from Xu et al [11].

### Diabetes-related knowledge

The questionnaire for general diabetes knowledge was employed to determine the knowledge level of diabetes among participants [12]. Eleven questions about diabetes were used to measure diabetes-related knowledge. Each question item had a scale response that ranged from 0 to 2 indicating "No", "Yes" or "Don't know". The total score on the scale ranged between 2 and 22. Internal consistency and reliability of the scale were assessed to be Cronbach's alpha ($\alpha$) = 0.697.

### Data analysis

Data were analyzed using R version 4.0.2 and IBM SPSS AMOS version 25. The normality of the distribution of numeric data was determined using the Kolmogorov–Smirnov. Categorical data were presented as frequencies. The Mann-Whitney test was used to compare skewed data. Data that were distributed normally were presented as mean ± standard deviation and a T-test was used to compare between groups. The correlation between diabetes-related knowledge, SCT constructs, and DSM was conducted with Spearman's correlation test. Structural equation modelling (SEM) was used to establish how the sample data closely fit the theory-driven model, by describing the relations of the dependency between the latent variables, which are usually accepted to have cause-and-effect outcomes [15]. A path analysis was conducted to describe the nature of the relationship between the HBM constructs and DSM, controlling for age, gender, and duration of disease. A $p$-value of less than 0.05 was deemed statistically significant.

## Results

The mean age of the study participants was 52.4 years (SD = 12.9) and a higher proportion of them were above 61 years (35.2%). There were more females than males (53.3% vs. 46.7%). Also, the male participants were significantly older compared to the females (54.2 ± 12.2 vs.

50.3 ± 13.3, p = 0.018). The average duration of DM was 9.9 years (SD = 6.9) and most of the participants have had the condition for 6–10 years (40.5%). Majority of the participants were self-employed (50.0%), married (72.4%), and had completed high school education (39.5%). Of the total participants, 73.1% (307/420) had T2DM with co-morbid hypertension, 65.5% (275/420) had uncontrolled FBG levels (> 6.1mmol/L) and 41.7% (175/420) had diabetes-related microvascular complications [Table 1].

**Table 1. Sociodemographic characteristics of study participants stratified by gender.**

| Variable | Total (n = 420) | Female (n = 194) | Male (n = 226) | P-value |
|---|---|---|---|---|
| **Age (years) mean±SD** | **52.9 ±13.4** | **51.1±13.7** | **54.2±12.9** | **0.018** |
| **Age groups (years)^** | | | | |
| 30–40 | 107 (25.5) | 57 (29.4) | 50 (22.1) | 0.093 |
| 41–50 | 82 (19.5) | 42 (21.6) | 40(17.7) | 0.325 |
| 51–60 | 83 (19.8) | 29 (14.9) | 54(23.9) | **0.020** |
| ≥ 61 | 148 (35.2) | 66 (34.0) | 82 (36.3) | 0.610 |
| **Duration of T2DM (years) mean±SD** | 9.9±6.9 | 9.6±6.9 | 10.3±6.9 | 0.343 |
| **Duration of T2DM (years)^** | | | | |
| ≤ 5 | 133 (31.7) | 65(33.5) | 68 (30.1) | 0.529 |
| 6–10 | 170 (40.5) | 80 (41.2) | 90 (39.8) | 0.823 |
| >10 | 117 (27.8) | 49 (25.3) | 68 (30.1) | 0.276 |
| **Occupational Status** | | | | |
| Self-employed | 210 (50.0) | 102 (52.6) | 108 (47.8) | 0.435 |
| Government employee | 116(27.8) | 51 (26.3) | 65 (28.8) | 0.585 |
| Pensioners | 44(10.5) | 21(10.8) | 23(10.2) | 0.874 |
| Not working | 50 (11.9) | 20(10.3) | 30 (13.3) | 0.367 |
| **Marital status** | | | | |
| Single | 17 (4.1) | 11 (5.7) | 6 (2.7) | 0.142 |
| Married | 304 (72.4) | 144 (74.2) | 160(70.8) | 0.587 |
| Divorced | 19(4.5) | 7(3.6) | 12(5.3) | 4831 |
| Widowed | 80 (19.0) | 32 (16.5) | 48 (21.2) | 0.215 |
| **Educational level** | | | | |
| Basic School | 103 (24.5) | 46 (23.7) | 57 (25.2) | 0.734 |
| High School | 166(39.5) | 77 (29.7) | 89 (39.4) | 0.999 |
| Tertiary | 151 (36.0) | 71 (36.6) | 80 (35.4) | 0.919 |
| **Regular source of income** | | | | |
| **No** | 92(21.9) | 45(23.2) | 47(20.8) | 0.553 |
| **Yes** | 328(78.1) | 149(76.8) | 179(79.2) | 0.553 |
| **With hypertension comorbidity** | | | | |
| No | 113 (26.9) | 59 (30.4) | 54 (23.9) | 0.456 |
| Yes | 307 (73.1) | 135 (44.0) | 172 (76.1) | 0.456 |
| **Microvascular complications** | | | | |
| No | 245 (58.3) | 121 (62.4) | 124 (54.9) | 0.120 |
| Yes | 175 (41.7) | 73 (37.6) | 102 (45.1) | 0.120 |
| **Current (FBG) (mmol/L)** | 6.93±2.41 | 6.98±2.53 | 6.88±2.30 | 0.786 |
| Controlled (≤ 6.1mmol/L) | 145 (34.5) | 67 (34.5) | 78 (34.5) | 0.996 |
| Uncontrolled (> 6.1mmol/L) | 275 (65.5) | 127 (65.5) | 148 (65.5) | 0.996 |

* Values are presented as a proportion of participants with diabetes-related conditions that suffered from such ailment. P-value is presented for comparison between male and female participants, ^values are presented as mean± SD and corresponding P-value obtained from t-test. Values highlighted are considered statistically significant.

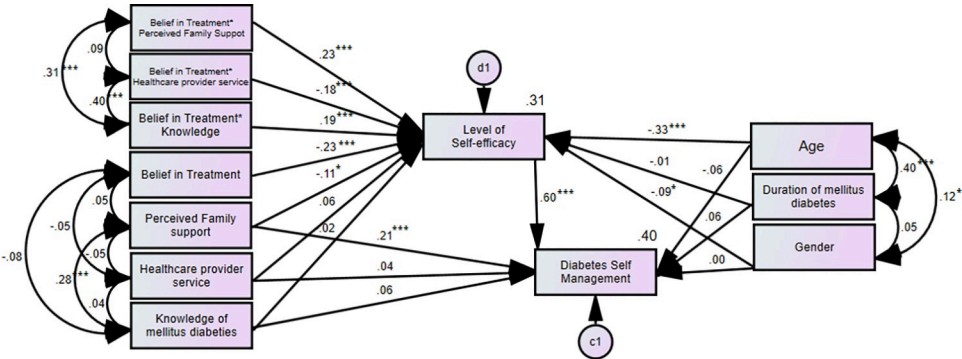

**Fig 2. Standardized estimates of path analysis model illustrating the perceived association between key determinants of diabetes self-management whilst controlling for age, gender, and duration of T2DM.**

## Strength and direction of association between study variables

The path analysis model revealed that age ($p = 0.176$), gender ($p = 0.901$), and duration of T2DM ($p = 0.119$) did not confound the relationships between the HPM constructs and DSM specified in the model. With respect to the associations among the control variables: significant positive associations between age and duration of T2DM (*Critical Ratio (CR) = 7.531*, $p < 0.001$) and between age and gender (*CR = 2.340, p = 0.019*). Among the exogenous variables, a significant positive association was found between knowledge of diabetes mellitus and perceived family support (*CR = 5.429, p < 0.001*). The associations among the endogenous variables revealed that level of self-efficacy had a significant positive relationship with diabetes self-management *(CR = 14.009, p < 0.001)*. In assessing direct significant associations between exogenous and endogenous variables, we found a significant negative association between perceived family support and level of self-efficacy (*CR = -2.545, p < 0.011*), and DSM practices (*CR = 5.279, p < 0.001*), supporting H3 [**Fig 2**]. No significant direct effect of knowledge of T2DM and healthcare provider service on DSM practices, implying that H1 and H2 were not supported. A significant association between the level of self-efficacy and the interaction between belief in the treatment and exogenous variables was observed [**Table 2**].

## Testing the fit of the conceptual model and evidence of mediated effect

The fit statistics indicate that good model fit was achieved (GFI = 0.976 > 0.95, AGFI = 0.943 > 0.9), with excellent parsimony-adjusted indexes (RMSEA < 0.024, 95% CI: [0.012, 0.043], PCLOSE = 0.642). The difference between the residuals of the sample covariance matrix and the hypothesized model indicates a good fit (SRMR = 0.044 < 0.080). With respect to the variance of the endogenous variables explained by exogenous variables, 31% of the variability in the level of self-efficacy was explained and 40% of the variability in DSM was explained.

In testing for the evidence of mediated-moderation, the simple slopes for the exogenous variables were tested on the level of self-efficacy at three different levels of belief in treatment effectiveness using the standard pick-a-point approach. We evaluated the significance of estimates of total effects (which evaluates the combined indirect and direct effects,) and that of specific paths through the mediation variable (level of self-efficacy), moderated by belief of treatment effectiveness, based on 2000 bootstrap estimates from the bias-corrected percentile method [**Table 3**]. This allowed for the construction of confidence bounds around the estimates obtained for conditional indirect and direct effects. There were significant differences in

**Table 2. Estimates (95% confidence intervals) of the total effect of exogenous, control, moderated, and mediation variables on DSM.**

| Total effect | Estimate (SE) | Critical Ratio (CR) | p-value | Remarks |
|---|---|---|---|---|
| *Diabetes Self-Management* | | | | |
| Knowledge of mellitus diabetes | 0.366 (0.260) | 1.409 | 0.159 | H1 not supported |
| Perceived Family support | 0.269 (0.051) | 5.279 | **<0.001** | H3 supported |
| Healthcare Provider service | 0.245 (0.246) | 0.996 | 0.319 | H2 not supported |
| Level of self-efficacy | 1.046 (0.072) | 14.607 | **<0.001** | |
| Age | -0.041 (0.030) | -1.352 | 0.176 | |
| Duration of mellitus diabetes | 0.086 (0.055) | 1.560 | 0.119 | |
| Gender | 0.089 (0.715) | 0.125 | 0.901 | |
| *Level of self-efficacy* | | | | |
| Knowledge of mellitus diabetes | -0.088 (0.160) | -0.550 | 0.582 | |
| Perceived Family support | -0.079 (0.031) | -2.545 | **0.011** | |
| Healthcare Provider service | 0.219 (0.150) | 1.458 | 0.145 | |
| *Belief in treatment effectiveness* | -0.391 (0.070) | -5.618 | **<0.001** | |
| *Moderator interaction with exogenous variables* | | | | |
| Knowledge of mellitus diabetes | 0.021 (0.005) | 4.151 | **<0.001** | |
| Perceived Family support | 0.006 (0.001) | 5.485 | **<0.001** | |
| Healthcare Provider service | -0.013 (0.003) | -4.037 | **<0.001** | |
| Age | -0.133 (0.018) | -7.509 | **<0.001** | |
| Duration of mellitus diabetes | -0.004 (0.034) | -0.130 | 0.897 | |
| Gender | -0.926 (0.436) | -2.124 | **0.034** | |

simple slopes (SS) for knowledge of T2DM on level of self-efficacy at the medium *(SS = 0.527, 95% CI: [0.175, 0.919], p = 0.005)* and high levels of belief in treatment effectiveness *(SS = 0.158, 95% CI: [-0.004, 0.323], p = 0.055)*. Similar results were found for perceived family support on level of self-efficacy at medium *(SS = 0.090, 95% CI: [0.025, 0.164], p = 0.007)* and

**Table 3. Evidence of moderated mediation for the relationships between diabetes self-management and knowledge of diabetes mellitus, perceived. family support, and healthcare provider service.**

| Exogenous variables | Parameters | Simple slopes for heat exposure on adaptation strategies (SS) | | | Conditional indirect effects (CIE) | | | Index of moderation mediation (IMM) | |
|---|---|---|---|---|---|---|---|---|---|
| | | Belief in treatment effectiveness | | | Belief inf treatment effectiveness | | | Moderated-mediation (MM) | |
| | | Low | Medium | High | Low | Medium | High | Index of MM | Remarks |
| **Knowledge of mellitus diabetes** | Estimate | 0.460 | 0.527 | 0.594 | 0.481 | 0.551 | 0.621 | -0.092 | No moderated-mediation effect |
| | 95% CI | [-0.124, 1.145] | [0.175, 0.919] | [0.191, 0.982] | [-0.114, 1.280] | [0.178, 1.014] | [0.198, 1.075] | [-3.746, 4.086] | |
| | p-value | 0.126 | **0.005** | **0.004** | 0.118 | **0.004** | **0.004** | 0.984 | |
| **Perceived Family support** | Estimate | 0.072 | 0.090 | 0.109 | 0.075 | 0.095 | 0.114 | -0.082 | No moderated-mediation effect |
| | 95% CI | [-0.024, 0.160] | [0.025, 0.164] | [0.022, 0.201] | [-0.024, 0.173] | [0.026, 0.177] | [0.025, 0.219] | [-0.716, 0.447] | |
| | p-value | 0.150 | **0.007** | **0.018** | 0.153 | **0.006** | **0.016** | 0.719 | |
| **Healthcare Provider service** | Estimate | -0.108 | -0.148 | -0.188 | -0.113 | -0.155 | -0.197 | 0.229 | No moderated-mediation effect |
| | 95% CI | [-0.525, 0.159] | [-0.432, 0.103] | [-0.504. 0.149] | [-0.559, 0.169] | [-0.452, 0.107] | [-0.512, 0.166] | [-2.021, 1.913] | |
| | p-value | 0.439 | 0.253 | 0.255 | 0.442 | 0.250 | 0.261 | 0.787 | |

Bold numbers indicate significant p-values and italic numbers indicate a trend toward significance.

high levels of belief in treatment efficacy *(SS = 0.109, 95% CI: [0.022, 0.201], p = 0.018)*. No significant difference was found in how healthcare provider service affects the level of self-efficacy irrespective of the belief in treatment effectiveness.

There were significant conditional indirect effects (CIE) for knowledge of T2DM at medium and high levels of belief in treatment effectiveness *(p < 0.05)*. However, the indexes of mediated-moderation (IMM). indicated that there was no evidence of moderated-mediation for knowledge of diabetes (IMM = -0.092, 95% CI: [-3.746, 4.086], p = 0.984) on diabetes self-management [**Table 3**]. We also found significant conditional indirect effects for perceived family support at medium and high levels of belief in treatment effectiveness *(p < 0.05)*, but no moderated-mediation effect was observed (IMM = -0.082, 95% CI: [-0.716, 0.447], p = 0.719). No significant conditional indirect effect and moderated-mediated effect was observed for healthcare provider service *(p > 0.05)* [**Table 3**].

## Discussion

The purpose of this study was to test a hypothesized model describing the effects of personal and environmental factors on DSM. We employed the path analysis model to allow for the identification of mediation influences (direct and indirect effects of variables) or factors and to demonstrate associations between the SCT constructs (exogenous and endogenous variables) and DSM. Based on our conceptual model, the present study assessed self-efficacy as a mediating influence in the association between personal and environmental traits, and DSM, and how belief in treatment effectiveness moderates this relationship. Altogether, the path analysis model demonstrated a moderate positive relationship between the SCT constructs and DSM. Our results showed that perceived family support had a significant direct effect on DSM practices and the combined path from perceived family support and level of self-efficacy had a significant effect on DSM.

The result of the current study also revealed that 31% of the variability in the level of self-efficacy by the exogenous variables and their interaction with the moderating variables. Level of self-efficacy expresses people's self-belief in their abilities to perform specific behaviours under a particular situation [39, 40] and in this present study, the conditional indirect effect of perceived family support and knowledge of T2DM via a level of self-efficacy was significant for patient DSM practices. This finding is in line with reports from a previous cross-sectional study by Didarloo et al., among Iranian women with T2DM [35]. In that study, they identified self-efficacy as the strongest predictor of self-care behaviours [35]. Another cross-sectional study by Tol et al [41] further reported that self-efficacy also had a greater influence on DSM practices among T2DM patients in a Thailand population. Systematic review investigations on the level of self-efficacy in patients with diabetes indicate that self-efficacy can positively influence health care behaviours [42]. Per previous reports and findings, medical researchers and health professionals have suggested that diabetes is a self-management disease and hence, it is the duty of the patients in part to take care of themselves [43, 44]. Recent pieces of evidence across multiple behaviour domains have shown that increased self-efficacy is associated with improved health outcomes, especially in T2DM patients [31, 45, 46]. As such, healthcare professionals should design better intervention strategies focussed on normalizing patients' experiences and validating their subjective experiences, ultimately promoting their confidence, and reinforcing patient self-efficacy.

Our study did not find any statistically significant direct effect of healthcare provider-patient communication on DSM practices. This result is contrary to findings of many studies that have shown that belief in treatment effectiveness and healthcare provider support have a positive relationship with DSM practices [10, 11, 32, 40, 47]. Healthcare providers can

encourage patients to self-manage with the supply of compassionate, practical, and individualized support. Nevertheless, belief in treatment effectiveness played a role in the indirect influence of healthcare provider services via self-efficacy on DSM practices. Like other studies [30, 32, 48], beliefs in treatment effectiveness may have influenced DSM practices through a change in self-efficacy. This relationship has been well substantiated based on the findings that T2DM patients effectively follow diet plans and self-monitor blood sugar levels when they believe in the benefits of undertaking DSM behaviours appropriately [48]. We found that the higher belief in treatment effectiveness observed a high slope for the exogenous variables' score on level of self-efficacy.

Public health education is a significant component of T2DM management. Like in most studies, there has been a statistically significant association between educational level and proper health behavior practices [11, 24, 29, 49]. In Particular, T2DM patients with some prior level of diabetes knowledge were more likely to undertake DSM practices [24, 28, 50]. These documented findings were confirmed with a report from a randomized single-blind controlled study that assessed the educational effect on self-management among T2DM [51]. The authors found that a two-week follow-up after a diabetes education program significantly improved self-management among T2DM patients [51]. A notable finding of this current study was that the direct path from knowledge of T2DM to DSM was nonsignificant. Instead, knowledge affected DSM indirectly through belief in treatment effectiveness and self-efficacy. These findings concord with reports from previous studies that have shown that knowledge did not lead to behaviour change directly [11, 29] but affected DSM indirectly through endogenous mediating variables [11]. In our analysis, knowledge of diabetes was significant for and affected both endogenous mediating variables. Adequate knowledge is important to improve DSM practices, but an individual's belief in treatment effectiveness and self-efficacy might also be involved. It seems that knowledge is necessary but not sufficient alone for behaviour changes in DSM among diabetic patients.

In this present study, we also observed family support as a significant positive factor for DSM practices. It has been well documented in the literature, that modification of behaviours and management of oneself are burdensome for people living with T2DM [47, 52]. One way to relieve this burden and promote better health outcomes is by providing social support via families and friends [53], which is positively associated with greater psychological and physiological well-being and reduced risk of morbidity and mortality in many chronic diseases [54]. Our findings of a direct influence are consistent with reports from previous cross-sectional and family-based intervention studies [29, 55, 56] which found family support to be a predictor for patient compliance with DSM practices. Patients' confidence level is built by strong support from family, which results in more efficient self-management and improved disease management [11]. Appraisal and information effect can be obtained through social support and this offers coping strategies designed to assist patients in managing diabetes-related stress and changed daily routines [36, 57]. A cross-sectional study by Mayberry and Osborn [36] reported that relatives who demonstrated better self-care behaviours were the ones that were better informed about diabetes and had better social support. Thus, it is essential to improve a better understanding of disease and the relationships between patients and their family members in clinical settings in such a way as to foster positive and supportive behaviours.

Overall, the cumulative effects of knowledge of T2DM, self-efficacy, and belief in treatment effectiveness influenced diabetes self-management practices and this relationship was well supported by our model. Perceived family support also showed a significant direct effect on DSM practices in this population. A limitation of this study is the cross-sectional nature of the study limits the capacity to unveil the causal relationship between predictors and DSM practices. Nevertheless, this is the first study among T2DM in a Ghanaian population, where a plenary

description of factors associated with DSM practices has been extensively explored. The application of the SCT in this study has allowed for the identification of predictors of DSM and analysis of various moderating factors in DSM practices among diabetic patients by controlling for cofounders such as age, gender, and duration of disease. Additionally, the current findings add substantially to our understanding of the significant role SCT constructs play in DSM practices among Ghanaian T2DM patients. Considering the very devastating rate of morbidity and mortality associated with T2DM, more efforts will be required to augment mainstream clinical management approaches by appropriately developing behavioural change interventions with a focus on mediating factors that will positively influence patient self-care management behaviours and practices.

## Conclusion

Overall, our results provide a detailed appreciation of the interaction between personal and environmental factors and their effect on proper DSM practices among T2DM patients. Self-efficacy, belief in treatment effectiveness, and prior diabetes knowledge emerged as the most significant facilitating factors for proper DSM practices among T2DM patients. In this relationship, self-efficacy served as a significant mediating variable. Perceived family support also showed a significant direct effect on DSM practices in our studied population. Iinterventions targeted to address mediator roles such as patient self-efficacy, belief in treatment effectiveness, and family support may be required to improve proper DSM performance among T2DM patients. Other approaches that target behavioural factors for proper DSM practices in the wider population should be encouraged.

## Supporting information

**S1 Dataset. SPSS file of datasets used and/or analysed during the current study.**
(SAV)

**S1 Questionnaire. Association between constructs of social cognitive theory model and diabetes self-management among Ghanaian diabetic patients.**
(DOCX)

## Acknowledgments

The authors wish to express their profound gratitude to all the staff and study participants at the Diabetic Clinic Komfo Anokye Teaching Hospital who voluntarily participated in the research.

## Author Contributions

**Conceptualization:** Yaa Obirikorang, Emmanuel Acheampong, Enoch Odame Anto, Evans Asamoah Adu, Enoch Ofori Antwi, Emmanuella Nsenbah Acheampong, Christian Obirikorang.

**Data curation:** Yaa Obirikorang, Emmanuel Acheampong, John Taylor, Sylvester Yao Lokpo, Evans Asamoah Adu, Bernard Adutwum, Enoch Ofori Antwi, Emmanuella Nsenbah Acheampong, Christian Obirikorang.

**Formal analysis:** Yaa Obirikorang, Emmanuel Acheampong, Ebenezer Afrifa-Yamoah, Sylvester Yao Lokpo, Evans Asamoah Adu, Freeman Aidoo, Eddie-Williams Owiredu.

**Methodology:** Emmanuel Acheampong, Eric Adua, Linda Ahenkorah Fondjo, Evans Asamoah Adu, Bernard Adutwum, Enoch Ofori Antwi, Emmanuella Nsenbah Acheampong, Eddie-Williams Owiredu, Christian Obirikorang.

**Resources:** Bernard Adutwum, Enoch Ofori Antwi.

**Supervision:** Yaa Obirikorang, Enoch Odame Anto, Linda Ahenkorah Fondjo, Christian Obirikorang.

**Validation:** Bernard Adutwum.

**Writing – original draft:** Yaa Obirikorang, Emmanuel Acheampong, Enoch Odame Anto, Ebenezer Afrifa-Yamoah, Eric Adua, John Taylor, Evans Asamoah Adu, Enoch Ofori Antwi.

**Writing – review & editing:** Yaa Obirikorang, Emmanuel Acheampong, Enoch Odame Anto, Ebenezer Afrifa-Yamoah, Eric Adua, John Taylor, Linda Ahenkorah Fondjo, Sylvester Yao Lokpo, Evans Asamoah Adu, Emmanuella Nsenbah Acheampong, Michael Adu Gyamfi, Freeman Aidoo, Eddie-Williams Owiredu, Christian Obirikorang.

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
