## [Decision Letter · Decision Letter 0]

15 Aug 2021

 PGPH-D-21-00139 Nexus between health promotion model constructs and diabetes self-management among Ghanaian diabetic patients: A mediation modelling approach. PLOS Global Public Health

Dear Dr. Acheampong,

Thank you for submitting your manuscript to PLOS Global Public Health. After careful consideration, we feel that it has merit but does not fully meet PLOS Global Public Health’s publication criteria as it currently stands. Therefore, we invite you to submit a revised version of the manuscript that addresses the points raised during the review process.

We look forward to receiving your revised manuscript.

Kind regards,

Jin Mou, M.D., MSc., MPH, Ph.D

Academic Editor

Journal Requirements:

Additional Editor Comments (if provided):

Dear Emmanuel,

We have received comments and reports from two reviewers regarding your manuscript "Nexus between health promotion model constructs and diabetes self-management among Ghanaian diabetic patients: A mediation modelling approach". I am inviting you to carefully read the suggestions and make revisions.

Please find attached the two reviewers' comments/opinions. If your have further questions, please dont hesitate to contact me.

Best,

Jin

Reviewers' comments:

Reviewer's Responses to Questions

**Comments to the Author**

1. Does this manuscript meet PLOS Global Public Health’s publication criteria? Is the manuscript technically sound, and do the data support the conclusions? The manuscript must describe methodologically and ethically rigorous research with conclusions that are appropriately drawn based on the data presented.

Reviewer #1: Yes

Reviewer #2: Yes

2. Has the statistical analysis been performed appropriately and rigorously?

Reviewer #1: Yes

Reviewer #2: Yes

3. Have the authors made all data underlying the findings in their manuscript fully available (please refer to the Data Availability Statement at the start of the manuscript PDF file)?

Reviewer #1: Yes

Reviewer #2: Yes

4. Is the manuscript presented in an intelligible fashion and written in standard English?

Reviewer #1: No

Reviewer #2: Yes

5. Review Comments to the Author

Reviewer #1: The authors presented an original research. I did not see any evidence that the paper has been published elsewhere. The methods sections have some challenges and could be addressed. The standard of English can be improved.

Reviewer #2: This is a well-written paper and the subject matter is also of importance to the policy makers in ghana considering that NCDs are on the rise. however, the authors need to fix minor issues indicated not Manuscript (see attachment). Few studies in Ghana and the African continent were reviewed and reported in the introduction and discussion sections. The authors could augment the paper with more studies from the sub region.

6. PLOS authors have the option to publish the peer review history of their article (what does this mean?). If published, this will include your full peer review and any attached files.

**Do you want your identity to be public for this peer review?** For information about this choice, including consent withdrawal, please see our Privacy Policy.

Reviewer #1: **Yes: **Dr. Reginald Quansah

Reviewer #2: **Yes: **Eric Nsiah-Boateng

---

## [Decision Letter · Decision Letter 1]

15 Jun 2022

Nexus between constructs of social cognitive theory model and diabetes self-management among Ghanaian diabetic patients: A mediation modelling approach.

PGPH-D-21-00139R1

Dear Mr Acheampong,

We are pleased to inform you that your manuscript 'Nexus between constructs of social cognitive theory model and diabetes self-management among Ghanaian diabetic patients: A mediation modelling approach.' has been provisionally accepted for publication in PLOS Global Public Health.

Best regards,

Julia Robinson

Executive Editor

Reviewer Comments (if any, and for reference):

Reviewer's Responses to Questions

**Comments to the Author**

1. If the authors have adequately addressed your comments raised in a previous round of review and you feel that this manuscript is now acceptable for publication, you may indicate that here to bypass the “Comments to the Author” section, enter your conflict of interest statement in the “Confidential to Editor” section, and submit your "Accept" recommendation.

Reviewer #1: All comments have been addressed

2. Does this manuscript meet PLOS Global Public Health’s publication criteria? Is the manuscript technically sound, and do the data support the conclusions? The manuscript must describe methodologically and ethically rigorous research with conclusions that are appropriately drawn based on the data presented.

Reviewer #1: Yes

3. Has the statistical analysis been performed appropriately and rigorously?

Reviewer #1: Yes

4. Have the authors made all data underlying the findings in their manuscript fully available (please refer to the Data Availability Statement at the start of the manuscript PDF file)?

Reviewer #1: No

5. Is the manuscript presented in an intelligible fashion and written in standard English?

Reviewer #1: Yes

6. Review Comments to the Author

Reviewer #1: i am satisfied with authors' response

7. PLOS authors have the option to publish the peer review history of their article (what does this mean?). If published, this will include your full peer review and any attached files.

**Do you want your identity to be public for this peer review?** For information about this choice, including consent withdrawal, please see our Privacy Policy.

Reviewer #1: **Yes: **Reginald Quansah
